# Design and Implementation of a Virtual Sensor Network for Smart Waste Water Monitoring

**DOI:** 10.3390/s20020358

**Published:** 2020-01-08

**Authors:** Edmundo Guerra, Yolanda Bolea, Javier Gamiz, Antoni Grau

**Affiliations:** 1Automatic Control Department, BarcelonaTech, 08034 Barcelona, Spain; edmundo.guerra@upc.edu (E.G.); yolanda.bolea@upc.edu (Y.B.); 2Project Innovation Department, Aigues de Barcelona Inc., 08003 Barcelona, Spain; javier.gamiz@agbar.es

**Keywords:** robotics, multi-agent systems, water sampling, industrial UAV

## Abstract

Monitoring and analysis of open air basins is a critical task in waste water plant management. These tasks generally require sampling waters at several hard to access points, be it real time with multiparametric sensor probes, or retrieving water samples. Full automation of these processes would require deploying hundreds (if not thousands) of fixed sensors, unless the sensors can be translated. This work proposes the utilization of robotized unmanned aerial vehicle (UAV) platforms to work as a virtual high density sensor network, which could analyze in real time or capture samples depending on the robotic UAV equipment. To check the validity of the concept, an instance of the robotized UAV platform has been fully designed and implemented. A multi-agent system approach has been used (implemented over a Robot Operating System, ROS, middleware layer) to define a software architecture able to deal with the different problems, optimizing modularity of the software; in terms of hardware, the UAV platform has been designed and built, as a sample capturing probe. A description on the main features of the multi-agent system proposed, its architecture, and the behavior of several components is discussed. The experimental validation and performance evaluation of the system components has been performed independently for the sake of safety: autonomous flight performance has been tested on-site; the accuracy of the localization technologies deemed as deployable options has been evaluated in controlled flights; and the viability of the sample capture device designed and built has been experimentally tested.

## 1. Introduction

Changes in industrial scale processes have usually been introduced by the development of new technical innovations. These changes also affect management and supervisory tasks, that is, with improvements in productivity and efficiency due to new technologies applied in productive processes, comes the need to manage this increased production as well as new requirements to make it work. In industrial scale production operations, the human effort is being made redundant at an increasing speed as a result of robotics advancements. This mainly impacts the physical efforts, so there are still plenty of tasks where human work is required owing to the versatile nature. Even the most advanced systems are powerless when confronted with the sheer versatility of human capabilities, especially in relation to cognition and data processing. These advanced systems can beat the human at specific tasks with ease, but only if everything related to the task is properly accounted, modelled, or built to precise specifications, making it hard to change. In monitoring and process management areas, where human expertise accumulated over years can be hard to model or represent in mathematical way, the best results achieved by the advanced systems are produced by using them as specialized support tools [1]. In this context, these support tools/system solve specific tasks, leaving the high level tasks and decisions to human operators.

Notice that, even with the help of these systems, monitoring and supervisory roles usually present frequently repeated operations that can be automated using recent advances in robotics. The most recent revolution in autonomous robotics is the development of fully robotized vehicles, not only cars and other UGVs (unmanned ground vehicles), but also unmanned aerial vehicles (UAV). This field has advanced quickly in the last years thanks to emerging technologies in micro-electromechanical systems (MEMS), sensors, and power management and storage. These technologies have led to the development of a new generation of battery-powered multicopters, which have started to gain a foothold in several industries: logistics and delivery, surveying of construction site and agricultural areas, videography, and so on. It is worth noting that, usually, the UAVs operated under direct control or strict supervision of human operators, with limited autonomous capabilities being used. In the mentioned industries, the surveying field is where the applications of these technologies is growing faster [2], as they can exploit UAVs’ mobility, speed, and range, while being more affordable than piloted alternatives.

Monitoring and surveying operations require sampling of products, probably more than once for each for them, like in wastewater treatment plants. In a typical treatment plant, waters with different levels of pollution need to be tested frequently, following concrete specifications and normative sampling of different basins and tanks. These operations require capturing samples from different spots in-site, being a task that is cumbersome, frequent, and systematic, thus being the kind of procedure that could be automated. Given the open structure of an outdoor wastewater treatment plant, with open air tanks and basins, UAV could hypothetically reach most of the sampling points, with better accessibility than human operators. Then, being a known open environment, the navigation challenges and risks can be managed with current technologies [3,4].

It is worth noting how most of the state-of-the-art related with UAV and water management is focused in surveying and inspection scenarios relying mainly on imaging sensors. These kinds of technologies have been applied from basic identification and measurement of water masses, to more complex approaches. In the work of [5], a combination of sensors, including both RGBD (Red, Green, Blue and Depth) and infrared imaging and laser scanning, is used to build accurate geospatial representations of water masses and the surrounding environments and evaluated different approaches. Other works focus on bathymetric measurement, so they rely on advanced photogrammetric processing of multispectral image data, captured through autonomous flights with UAV platforms. In the work of [6], additional data are used, so to complete the bathymetric surveying, an underwater ROV (remotely-operate vehicle) was used to deploy a depthometer. Works of an experimental nature, where sensing of waters is done indirectly (such as its presence in vegetation), have also presented interesting developments when introducing autonomous technologies UAV, like the work of [7], where the impact of measuring water contents through UAV is compared to that of other approaches using satellite imaging data.

There are some works where the data sensing processes actually contact the waters to be analyzed. In the work of [8], a theoretical design for a UAV platform able to analyze waters is presented and discussed, using a fixed pulleys system to lower a multiparametric probe. This work was focused on some theoretical design aspects and developing the dynamical model of the UAV and testing it in simulation with a dynamic inversion controller. A more practical approach is describe in the work of [9], where a high specification automated UAV platform was presented to analyze maritime and oceanic waters, with constant monitoring and supervision of the autonomous flight on a large-sized platform. The system presents a sample capture device requiring human intervention for sample retrieval/device resetting, with a main function of deploying surface drifter sensors. Note that these kinds of platforms are generally of large dimensions, targeting scenarios where no human actors are expected to be present, like the work of [10], where a UAV platform is used to automatically perform irrigation and fumigation tasks, being able to maneuver with a 10 kg liquid tank as payload.

Note that the full automation of any task through UAV technologies requires that the robotized UAV platforms used present enough autonomy in their behavior to act and decide with little supervision in environments not fully known a priori. It is worth noting that, while it is possible to model a known industrial environment, the activity performed itself implies that there may be dynamic elements hard to model and represent. Thus, the robotized UAVs need to be able to act on their own agency to perform an a priori known task in a mostly modelled environment and conditions. To produce these kind of robotics systems, there are several engineering methods [11] and support technologies [12] available, with the multi-agent system (MAS) being one of the most successful. Under this paradigm, the different tasks or parts of the problem to solve are distributed to different agents, which focus on their own tasks. As each agent is a different process responsible for a task, they can work in parallel, dealing with different challenges of the system, ignoring the rest of the system and data. This simplifies each agent (as it only solves one task or problem), who only needs knowledge about its task (easing data processing and management operations), and who depends only on a small number of other agents, if any (isolating processes and execution threads). All these features lead to systems that are simpler to design, implement, and test, whereas the agents deal with smaller problems. They are easier to maintain, because as long as the interfaces remain the same, the implementation of any agent may be easily replaced thanks to the modularity of the approach. They are also more robust, because agents run independently, so failure of an agent does not necessarily mean failure of a system; only other agents dependent on the affected capabilities of the agent will be impacted. Then, in the case of failure of an agent, as each one presents its own state and decision-making capabilities, it will decide individually how to proceed after the error. An effective demonstration of how to use these capabilities to produce a robotics MAS using the ROS framework [13] can be seen in the work of [14]. In said work, MAS methods described in the work of [15] are used to produce a full solution to manage a system composed of an underwater surveying vehicle deployed from a surface vehicle, all controlled from a ground base station.

From a theoretical point of view, MAS methodologies are related with behavior-based systems, as the agents in a MAS can be designed according to behavior-based criteria, which allows to combine complex planned policies, as in deliberative control, with otherwise purely reactive patterns. For example, some research has demonstrated coordination and integration of reactive behaviors, as a key problem in autonomous navigational strategy for mobile robot, to avoid crashing with obstacles by turning to the proper angle while executing tasks in a partially known and complex configuration of obstacles. In reactive navigation, the fuzzy approach has been proven to be a commendable solution in dealing with certain ambiguous situation. As the complexity of robotic environment increases, fuzzy logic, which can handle infinite navigation situations with a finite set of rules, is implemented here for modeling uncertain systems by facilitating human-like common sense reasoning in decision-making in the absence of complete and precise information. In the work of [16], the kinematic architecture of a mobile robot model and attention is focused on the design, coordination, and fusion of the elementary behaviors of mobile robot based on weighting factors produced by fuzzy rule base taking into account current status of robot. In the work of [17], the authors consider the problem of autonomous exploration of unknown environments with single and multiple robots. This is a challenging task, with several potential applications. It is proposed that a simple, yet effective approach that combines a behavior-based navigation with an efficient data structure to store previously visited regions, allowing robots to safely navigate, disperse, and efficiently explore the environment.

In the work of [18], a mobile robot navigation issue by combining the ideas of behavior-based control and fuzzy logic control is presented. The learning and behavioral predictive control is sometimes proposed for efficiently planning autonomous real-time pre-specified trajectory tracking and obstacle avoidance, such as in the work of [19], applied to an omnidirectional wheeled robot using fuzzy inference algorithm. In the work of [20], a nice survey on multiagent systems (MAS) is presented dealing with behavior management in collections of several independent entities, or agents. In the work of [21], the schemas underlying robot behavior are analyzed in depth, showing the reciprocity between robotics and biology and sociology.

Recently, deep reinforcement learning (DRL) has shown great potential in mapless navigation of mobile robotics [22], where a hierarchical control framework is proposed to address this problem. An autonomous mobile robot has to possess a locomotion way suitable for the mobile environment and to move autonomously. In the work of [23], the research is aimed to operate an autonomous mobile robot by determining the control input from the features of acquired camera images. Some advantages of this method are the following points: there is no need for self-localization, so that the amount of computations is small, and it can be controlled by a human-like method. Apart from autonomous robots, in the work of [24], the authors develop an optimal control framework that takes the full-body dynamics of a humanoid robot into account through exploration of full-body dynamics in, especially, an online optimal control approach known as model predictive control (MPC).

In the authors’ previous work [25], a proposal for the architecture of a sampling system for a waste water treatment plant was presented. In that work, a single UAV worked jointly with an UGV to collect and store batches of samples to be delivered as a set, seeding some ideas and concepts used in the present work. In this paper, a new system is designed and partially tested, where a network of robotized UAVs, each of them with autonomous decision-taking capabilities to minimize risks, is used created a virtual “high density” sensor network, capable of sampling any surface point of the tanks and basins, as if thousands of sensors had been deployed. This sampling process can be performed with a newly designed and simplified sampling probe, or using a multiparametric sensor.

This work sets the basis of this new research, where a UAV is part of a sensor network with a newly designed sampling probe to take samples from any point in wastewater tanks and basins. The new system supports a network of UAV platforms, whose design architecture is presented as a contribution, like its role as a sensor in a network sensor, the new custom sampling probe, and several results obtained during the development and testing of a prototype UAV.

Thus, the following section of this paper presents a review of the main features on the sampling problem; a brief description of the study case environment considered; and an explanation about how the proposed system works, describing the operations at high level. After framing the problem and describing the operation of the proposed solution, the section “System and Hardware Structure” describes how the system was built, listing its main components, organization into different subsystems, and emphasizing in the actual hardware of the UAV prototype constructed. The UAV is described thoroughly, detailing the hardware integration and connection at the interface level. The next section, “System Design and Software Architecture”, shows the design produced using a multi-agent system point of view, describing software organization (at the logical level) and implementation level, with a discussion on certain virtual software components. After describing the problem to solve, and the solution proposed in terms of how it operates and how it is implemented a hardware and software level, the experimental results produced are discussed. Three main experimentation areas are considered: the designed sampling probe, which is validated experimentally off-board (i.e. not flying in the UAV); evaluation of the available localization technologies so that relevant points of sampling can be reached with accuracy; and finally, some limited early results from flight tests in an area adjacent to the target basins to sample. To conclude the work, the contributions and advantages of the proposed system are presented and discussed, also laying pending issues to solve and future lines of work.

## 2. Monitoring of Plants with a Virtual Sensor Network

### 2.1. Monitoring and Supervision of Wasterwater Treatment Processes

Monitoring and supervision tasks within a wastewater treatment plant require accurate sampling capabilities in order to properly estimate the different processes states and being able to control them properly. These actions have to guarantee compliance of the legal requirements for the different metrics to be accounted in the resultant water. The indicators usually considered include from chemicals properties, like for example pH, to concentrations of specific materials/pollutants (both organic and inorganic in nature). In many cases, these processes require obtaining multiple samples with specific sampling procedures (nitric nitrogen samples have time limits to perform the analysis, generally under two hours, while microbiological test samples are to be kept away from direct sunlight) [26].

In any case, all sampling processes are designed so that they produce samples as representative as possible with respect to the original, so that the measurements are significant. In small scale analytics, this generally implies shaking/stirring/centrifugation the sample, or any similar process that can improve the homogeneity of the body to be sampled. These techniques can be used in limited ways when dealing with large infrastructure, like basins and tanks; and as the transport processes present slow dynamics in them, low flow velocities drag the dilution processes all along the basins. Still, the level of homogenization produced by these processes is relatively low, and it comes at the price of varying metrics along the tanks as the slow dynamics make it behave like a buffer, which introduces delays, which are challenging to model. In turn, this means that there will be waters subjected to varying degrees of treatment at each step, subject to hard to model disturbances even before considering that waters can enter a basin at different times, with distinct pollution levels.

All these particularities about the sampling process for large basins mean that the only way to guarantee representative samples is to repeat the process at several points in a given basin. Notice that it is entirely possible that the most convenient sampling points with respect to the measurement process may lay in hard to reach areas, with complex safety requirements and/or regulations to allow human workers to work in the area. If a set of known sampling points can produce a model that satisfies the analytics requirements for representativeness and accuracy, fixed sensors can be installed into the basins, but these kind of sensor network solutions present several drawbacks, most of them related to the lack of flexibility; any change that would require varying the sampling point reduces the value of the sensor network unless updated, which is usually expensive. 

If the sampling process is performed by human workers, there are still additional challenges and risks to consider beyond the accessibility to the work points to procure the samples: specific analytic requirements (as commented before, sun exposure, shelf-life of the sample), weather and adverse climatology, special industry-related risks like toxic substances or gasses, and so on. While all these risks can be managed with proper methodologies and equipment, its cost can become burdensome, and at the end of the day, they are still present if there are human workers; risks can be minimized through management, but cannot be eliminated if the operations are still physically performed by said workers. This simple fact acts as a value multiplier for any benefit provided by task automation as it removes risk for humans from the scenario.

### 2.2. Vritual Sensor Network using Roboticed UAVs

To remove most of the risks associated with the sampling tasks performed by human operators, a sensor network was designed that allows flexible representative sampling in most areas of the basins. Note that achieving flexible sampling capabilities with a conventional sensors network would not be possible, as the sensors are fixed. The best alternative under a conventional network architecture would be deploying a high density network of sensors, with hundreds of sensing devices per basin/tanks, which would be considered unfeasible from an economical point of view. This is the requisite conventionally provided by human workers, flexibility, in this case, being able to deploy the sensor in a variable position. In order to build a feasible sensor network, which does not rely in human workers for sensor deployment, a virtual sensor network was designed. 

The proposal presented allows solving sampling tasks in environments with open basins using a high density virtual sensor network. The system is actually built in the form of a network of robotized UAVs with autonomous capabilities to deploy sensing devices/capture samples. This way, they can sample almost any point in a given open air basin, acting as if hundreds of sensors/sampling point were installed in each basin. Two different main modes of operation can be used, depending on the requirements of the sampling process, that is, the UAVs can either capture and retrieve a sample in a given point, or perform a measure in real-time in the given point. In order to achieve this, each component UAV in the network can be fitted with two different sensing equipments: a multiparametric sensor probe (which analyses the waters in real-time) or a designed sample capture probe, which can retrieve and hold up to ~400 cm^3^. The device in each UAV is attached below it and connected to the robotics system, enabling quick changing of configurations if required. This means that UAVs will required specific landing platforms based on what type of sensor is deploying.

Under normal operation, a human worker responsible for managing the sampling policy sets a list of sampling tasks. A given sampling task indicates which basin has to be sampled, which area/point of it, and any other requirement or instruction relative to the procedure. This includes which metrics must be analysed, any restriction or requirement (at which time the sample has to be produced, restrictions on handling the sample, keep the source sample for validation analysis, and so on), and a priority. With these data, the system checks the available sampling UAVs, and tries to plan a set of sampling missions (a sampling solution), which solves the maximum number of sampling tasks requested by the human worker. This plan considers real-time data from the UAV platform, thus considering the predicted availability of platforms (as they must recharge between sampling missions).

Each sampling mission starts using a combined height map and accessibility map to define a trajectory from the take-off point to the sampling point, avoiding all possible obstacles by performing a square signal-like trajectory. During the take-off, a minimal height that guarantees clearing the obstacles is reached; for example, for the considered plant, the highest obstacle is below 13 m, so a height of 16 m would be enough considering possible errors in positioning and disturbances. Once this height is reached, the UAV travels in a secure trajectory (usually a straight line, but it is possible to define inaccessible areas in the map that will be avoided), maintaining said height until the sampling point. Once at the sampling point, the UAV will decrease its height, until reaching a point over the water surface defined according to the sampling procedure. After this, the UAV is to remain as close to the stationary set point as possible for a set time (needed for the multiparametric sensors to set on the different values, and for the sample capture probe to be filled). Once this time passes, the UAV will gain height again to the safe altitude considered earlier, and will travel back, generally reversing the trajectory to the sampling point, as in most cases, it can be assumed that, at the start at the mission, it took-off from the relevant landing pad.

For a given mission where there is no need to do laboratory work, requirements to keep the samples, nor performing later validation tests, the measurements will be obtained with the multiparametric sensor probe. This means that a UAV deploying the probe will be sent to the designated measurement point, hover over water at a distance that enables submersion of the probe, and the return back. Notice that, in these cases, the results can be relayed immediately using the 4G/WiFi capabilities of the communications subsystem (Figure 1), enabling it to perform several continuous measurement operations under certain circumstances; the UAV has enough battery left to guarantee safety of operations, and the next measurement requirements in terms of analytic performance are met by a *used* probe (as opposed to a clean/just serviced one).

If the multiparametric probe is not suitable for the sampling mission (the sample must rest in laboratory under controlled for five days, like BOD5 metrics, or the multiparameter sensor cannot satisfy any of the required analyses), a UAV robotized platform with the sample collector probe will be dispatched in place. Operating this kind of UAV presents additional challenges given that the weight of the payload and its distribution will vary during flight as result of the filling and transport of the wastewater sample. The sample capture process required flying over the basin, at a low altitude over the water surface, so that the probe, which is attached rigidly solidary to the UAV, is submerged into the basin. The probe needs to be submerged for a few seconds (it is filled in under 15 s), and after being filled, the flotation valve will close the probe, the UAV can fly up, and then proceed to a delivery station. While hovering over this delivery station, the robotics system will release a magnetic valve sustaining the probe, so that it falls into the relevant point. Alternatively, a specially built landing pad can be used, so that the UAV platform can land with space for the probe (Figure 2).

## 3. Hardware Description and Integration

To validate the system, a prototype robotized UAV was designed and built (Figure 3), largely based in concepts from the work of [25]. The prototype built would operate as one of the UAV network nodes composing the high-density virtual sensor network. In order to ease operation and maintenance, one of the key design criteria was employing off-the-shelf (TOS) technologies for hardware and open source software, thus reducing costs. Notice that, at first sight, buying a fully integrated commercial UAV could look a better option, until several aspects are considered. First of all, cost competitive UAVs are usually mass produced as part of closed systems, thus there is no option to customize the flight management unit (FMU) controllers to support dynamical variations of the payload mass. Although there are commercial platforms developed for research (mainly for developed for the perception field), and thus based on open FMU, they would not provide competitive pricing once the cost of modifications to fit the required custom hardware was accounted for. The rest of the system would be composed of additional sensor nodes (i.e., other UAV platforms) and the control system, which can be deployed in an ordinary commercial PC.

The UAV itself is a X4 configuration platform [27], managed with a PixHawk 2.4.8 FMU using the PX4 flight stack [28]. The power system was modified with a custom board to admit two 6S 10Ah batteries. An EGNOS-enabled (European Geostationary Navigation Overlay System) Ublox GPS (Global Positioning System) and a laser altimeter are used as sensors to the avionics system connected to the FMU. To manage and validate the avionics independently, a radio controller at 2.4 GHz and a telemetry link at 915 MHz were also integrated, enabling manual control if required during testing.

While the FMU introduced is enough to provide basic autonomous flight capabilities, an additional Odroid XU4 single board computer (SBC) was deployed to dote the UAV with additional capabilities. Several additional sensors were connected to the device, namely, a USB camera, a set of MaxBotix ultrasonic collision sensors, and a network beacon transceiver. This set of sensors connected to the SBC allows fully autonomous flight and operations in known environments, and the GPIO/USB (General Parallel Input/Output) are used to integrate the different sampling systems; the GPIO was used to control the magnetic valve of the sample capture probe, while a USB interface can be used to connect commercial multiparametric sensors. The complete hardware integration architecture can be seen in Figure 4.

To enhance the modularity and maintainability of the system, different organization approaches were combined. Firstly, all the components were grouped into a set of subsystems, including all the hardware and software related with the relevant purview. Notice that the subsystem division relates mainly to the specific tasks or jobs performed by the hardware. Below this subsystems level, the software components were designed to be organized according to a multi-agent system (MAS) architecture, largely based on the methods detailed in the work of [14]. Using a MAS approach was considered the most convenient option, as it essentially introduces an abstraction layer, which allows optimizing modularity within the system. The software design unit from a MAS point of view is the agent, which generally describes a process or set of tightly coupled processes, which are responsible for a given task having agency over it of its own. Thus, these components present a high level of abstraction from a design point of view, but from an integration point of view, they operate as ordinary software, and thus operation over the hardware with a given set of lower-level libraries and drives, which acts as the interface layer. Note that exploiting the modularity potential of a MAS architecture requires a clear definition of responsibilities of each agent, as unlike other behaviour-based robotic systems [21], the tasks in the designed platform present very low orthogonality, that is, most of the processes that could be coded as behaviours are heavily dependent on the same inputs and output, interfering with each other’s functions. This fact is mainly because of the restrictions imposed by the flying nature of the UAV, which translate into a need for centralization of several functions in order to ensure the real-time performance and validation required to operate it safely.

This multi-agent architecture is implemented as a MAS framework over the ROS middleware. ROS is a meta-operating system, which provides most of the interface layer software and many components to simplify building a network of agent components, which can be easily mapped as ROS “nodes”, its basic execution unit. An ROS node can be configured (statically and dynamically); instanced several times across different devices in the same ROS network; and its execution can be managed according its own algorithms (i.e., its own agency), although it can also be externally supervised. So, for a given subsystem considered in the first organization level, a set of agents will be implemented (most of them as ROS nodes), acting to solve a specific problem or task providing a high level function, establishing communications between them when required.

As commented earlier, software design under a MAS approach emphasizes modularity, which is supported at the implementation level by the middleware part of ROS. This translates directly into greater robustness, because, if designed correctly, each agent should be capable of an initial level of self-supervision, and implement the necessary procedures in failure cases; and maintainability, such as swapping a given agent/node for another different (e.g., a new perception algorithm for the same sensors), should be a simple task as long as the role/tasks and interfaces specified are respected. This is further supported by the ability to reconfigure the nodes even during operation, meaning that the same code can be deployed easily in multiple UAVs, booted with generic parametrization, and can be reconfigured to the needs of the sampling system during its pre-operation steps. This implies that several agents (and also subsystems) will be present in multiple components of the sampling network, that is, are required at each UAV that could eventually be deployed. On the other hand, several subsystems will be present only as part of the whole system (not a node of the sampling network), and operate with a single instance in mind.

The subsystem working as a single instance deals mainly with infrastructure coordination and management tasks:
Main control subsystem: This subsystem is composed of a set of ground stations to supervise the avionics of the UAVs and a control and supervision station for the sampling network. These machines run the central master agent and all the other agents required to coordinate the sampling operations of the different UAVs. They also deploy the user agent, and those required to manage infrastructure, like the beacon network agent. The development efforts were focused in producing a robust and easy to use API layer, which can be used as a foundation for any desired GUI/UX design.Beacon network subsystem: It is composed of a beacon sensor network used to help with UAV localization. The beacons are based in the CC2530 [29] chip from Texas Instruments, being a cheap alternative to other sensor network-based technologies for localization.Communications subsystem: The communications are implemented using multiple wireless channels. The radio telemetry from each of the FMU is observable through a ground station, while an ROS network is run over 4G through a VPN based on OpenVPN, enabling full communication through ROS, including compressed image streaming. Note that the communication system is considered as a single instance, though it presents components in all of the platforms.

The prototype sample collector UAV (and any other platform added in the future) presents a set of subsystems related to the flight functions, robotics operations, and the customization of the power supplying hardware components:
Robotics control subsystem: This subsystem is composed of the Odroid XU4 SBC and the sensors and devices connected to it (including the multiparametric sensor and the sampling probe). The main responsibility of this subsystem is to provide the capabilities required to turn a conventional UAV into an autonomous platform capable of automated sampling.Avionics subsystem: It is composed of all the hardware and software required for basic flight operations, including a PixHawk FMU, ESCs (electronic speed controls), propeller, GPS sensor, and landing gear controller and servos.Power subsystem: This system provides power to all the devices on-board a UAV, including the avionics and robotics subsystems. This is done with a custom board, which implements a balanced charger with galvanic switching between two batteries, allowing to isolate them to avoid damaging them by overdrawing. The same board also produces the different voltages required by the components of the system (23 V, 12 V, and 5 V).

## 4. Software Design for ROS under an MAS Architecture

The mentioned ROS framework used to implement the interface layer and support the MAS paradigm is based in ROS Kinetic Kame distribution. This software is deployed over Ubuntu 16.10 on an Odroid XU4 SBC (for the prototype sample collector UAV) or a conventional PC. The MAS paradigm provides not only the design principles to specify the set of agents needed, and guarantees their modularity and robustness, but also a useful abstraction level to manage the software. This translates into a virtual infrastructure to organize and design high level capabilities, including those not specifically designed or built explicitly within them. These include several software components and capabilities provided at the FMU, which, although they are not build as ROS nodes, are considered as agents (given their levels of self-agency). These agents are related or compose part of the avionics subsystem, and as such, they have to guarantee the flight capabilities and stability, so they run independently in the FMU, while the SBC will deal with high level perception and automation tasks, which generally are not considered as critical. In a robotized UAV, the SBC and the FMU can connect through several interfaces, and for the prototype built, a serialized port was used over Odroid’s GPIO.

The main weakness of using an ROS framework as a foundation to build the designed MAS is its reliance of a “master node”. This node essentially provides a name resolution service, being the “master agent”, and helps during the routing and initial steps of every agent to agent connection, providing also a unified MAS level clock when starting each agent. This means that ROS-based systems generally are centralized, proving a critical point of failure, especially when dealing with mobile autonomous robots, like the automated UAV, as not even the connection with this centralized node can be guaranteed at all times, irrespective of the network layer technologies used. Deployment of the master agent into a sample collector UAV would guarantee the integrity of its operation, at the price of compromising the rest of the system, while deployment within the hardware running the main control subsystem would jeopardize all the UAV platforms. This issue was solved using the multimaster_FKIE library [30], which enables utilization of several master agents/nodes in a single network. Thus, as described later, the main control subsystem presents and “central master node”, which is used to ensure periodic synchronization and communication at whole-system level, while each robotics control subsystem present at the UAV level will present a “local master agent”. This master agent enables operation of the robotics control subsystem for the UAV where it is deployed. This way, loss of connectivity between the main control subsystem and a given robotics subsystem does not disrupt the local MAS/reliant subsystem in the UAV. The same multimaster_FKIE toolset provides the synchronization libraries to ensure system integrity with respect to the connected systems; this is achieved by restricting synchronization of each local master agent to the central master agent, avoiding excess bandwidth consumption and data propagation loops that could be caused by UAV to UAV synchronization.

Thus, the agent level software used in the system can be grouped based on the execution environment. Firstly, we have those agents running in the PC stations of the main control subsystem:
Central master agent: This agent coordinates the MAS implemented over the ROS framework from the main control subsystem, thus providing access to the whole MAS through the local master agents.User agent: This agent provides the user interfaces to monitor, supervise, and control the whole system.Sampling network manager: Data on the network sampling capabilities and requirements are managed by the agent that will be responsible for deciding which sample collector UAV platform will deal with each specific task, finding the best sampling plan solution given the requirements of the operator.Central logger agent: This agent provides the logging services for the ground stations, user agents, and all the messages relayed through the main control subsystem. Note that it may present redundancies with respect to data stored in local logger agents to assure that data loss will be minimal in case of critical failures.Beacon network agent: This agent supervises the deployed beacon network to allow accurate localization when the SBAS (satellite based augmentation system) enhancement is unavailable.

Secondly, we have those agents related with the robotics control subsystem, or closely related to it, which are executed in the SBC:
Local master agent: This agent coordinates the multi-agent system implemented over the ROS framework in each sample collector UAV. It provides the necessary tools to support the local part of the MAS (contained in the UAV) and establishes the communication with the rest of the system through synchronization with the central master agent when the connection is available.Mission management agent: The parameters of each trip performed by a sample collector UAV are received and kept by the mission management agent, allowing the robotics subsystem to order the avionics subsystem to flight in order to collect the sample. This agent also implements decision-making policies with respect to some partial failures of the robotics subsystem during flight. Its behaviour is codified so that it prioritizes stability, obstacle avoidance, integrity of the platform, and complying the mission, in this order. This means that any circumstance notified by the agents responsible for that task can take priority over the mission, for example, if the power management agent notifies a battery failure/loss of power, this agent will decide, according its designed behaviour, to message the navigation agent with a return to landing pad (or even a land as soon as possible) notification.Navigation agent: The navigation agent uses inputs from localization agent and the UT (ultrasound) collision agent to compute the trajectory to make the UAV fly according to the mission management agent parameters and the height and accessibility map. This agent also codifies a failsafe behaviour should the mission management agent become unresponsive. Should that note fall and fail to respawn, the navigation agent will start a return to landing pad sequence.Localization agent: Data from the agents of the different sensors available at the UAV are synchronized and fused into an IEKF (invariant extended Kalman filter) to estimate the localization of the UAV with as much accuracy as possible. For accurate landing and positioning on fixed known environments, a fiducial marker-based estimator is available. This agent can reject data from any sensory input in case it became unreliable according to statistical criteria, and switches between the estimation modes available according to the part of the mission notified by the mission management agent. If a return to landing pad or land as soon as possible signal is notified, the localization agent is responsible for finding a clear space in the occupancy map to have a prepared emergency landing spot in the case it was needed.Camera agent: This ROS level driver interfaces with the low-level camera driver and publishes the data from the camera into the ROS framework so other components of the MAS can use it. Note that this agent is also responsible to process images when required, and compress them if they are going to be used beyond the reach of the local ROS framework of the UAV.UT collision agent: The low-level driver for the UT collision detector interfaces with this agent so that the data can be processed and turned into useful inputs for the navigation process at the relevant agent. In the case that UT sensor detects any object within a given threshold (usually under 1.3 m), an initial danger signal is sent to the navigation agent, while once a lower bound is crossed (around 0.8 m), a high priority avoid collision signal is sent to both the navigation and the mission management agents.DGPS agent: This agent processes the data from the DGPS sensor deployed for ground-truth computation/validation purposes when available. Making these data available in the MAS through the ROS framework allows to record it synchronized into the relevant logger agent, and to be post-processed for validation of the localization and navigation processes.Beacon network sensor agent: This agent provides localization estimation through trilateration based on the nodes of a deployed known network of beacons. This localization method is designed as a fall-back in case of SBAS shortages, which can disrupt the localization of the avionics agent.Multiparametric sensor agent: This custom ROS level driver interfaces with the multiparametric sensor API to set its initialization parameters, initialize and manage the sampling process, and publish its data in the local ROS framework, so it is available for the mission management agent.Sampling device agent: This agent interfaces with the GPIO of the SBC Odroid to manage the low-level controller hardware of the custom-built sampling device.Local logger agent: The local logger, instanced at the sample collector UAV level, allows recording data locally, useful to capture data from high bandwidth sensors to test and validate algorithms out of flight or off-board.

The last group of agents is executed in the FMU, and thus are parts of the flight stack PX4 [28]. Notice that their description as agents is mainly virtual, as entities with specific tasks, resources, and their own agencies and procedures defined; and although they are part of the same software stack, they present their own interfaces in the ROS framework under the guise of their respective topics and services:Avionics agent: The avionics agent runs in the FMU, where it manages the sensors connected to the FMU, the basic flight modes, and some emergency protocols in case the robotics subsystem becomes unstable/unresponsive/unavailable and emergency manoeuvers are required for safety.Power management agent: This agent is distributed between the FMU and SBC, in order to properly monitor the energy consumption. At the FMU level, it can trigger reactive behaviour in response to power loss, while at the SBC level, its output is received by the mission management agent.Low level control agent: This agent implements the low-level controllers to manage the ESC and motors of the UAV, thus managing the propulsion part of the avionics subsystem.Landing gear agent: The landing gear agent manages control of the hardware to fold the landing gear during flight and prepare landing operations. In emergency situations or safe modes, it will be set to prepare to land or to avoid a crash.

So, even being only virtual agents, from a design point of view, they are part of the multi-agent system, and will complete the MAS architecture, seen in Figure 5.

In terms of communications, the ROS framework provides the agents of the MAS with two different paradigms: publisher–subscriber, known as topics; and server–client, which can be implemented as services or actions, depending on duration of the task/request or other conditions (see Figure 6). Notice that even the agents that are implemented in the FMU, which are not coded as conventional “nodes”, present the same communication capabilities in the ROS framework as they can interface through the MavROS library. So agents will present an active life cycle during which they coordinate and communicate to perform tasks, starting with the launch of a node (using ROS terminology), when the agents (or different instances of an agent present at several UAVs) are configured and the execution is started. During this phase, all the services/actions and topic publishers (preferably also the subscribers) are declared, and registered in the relevant master agent. Data of the relevant topics and services are then shared between the local master agents and the central master agent, so the sensing network system can operate correctly.

After these initial steps, each agent may present different behaviors, namely the following:
Single use behaviour: a set of actions to be performed a single time; used commonly to complete configurations and adjustments relying on the fact that other agents of the system have been initialized successfully (e.g., define the initial take off point, capture metadata on sensor configurations).Timer behaviour: a set of actions to be performed after a given time has passed.Cyclic behaviour: one or more actions to be performed cyclically (reading sensors, estimating pose, commanding the UAV platform, and so on)Event-based behaviour: a set of actions triggered by an event.

The event triggering and specific behaviour may be of different kinds:
incidences detected in an ordinary communication topic, that is, specific messages received or lack of received messages for a given time;notification of a service request or an action request;removal of necessary topics for correct/normal behaviour;removal of required service/action server that is normally used.

The event triggering described by the two first cases will be part of regular functioning in most cases, as it is common to include control message in topics (though it is recommended to use services to verify reception), so it is to find spurious patterns in sensor data that have to be filtered. This may also lead to a lack of messages from sensor-processing agents towards higher level agents, and these cases are considered and accounted for. On the other hand, the two other cases can pertain to normal operations or to error conditions. When topics or services are removed as part of its operation, it means that the agent has reached its end of life phase and is closing connections, while if they become unavailable without proper notification to the master agents, it means that there has been an error/kill condition for the executable. The ROS framework offers the possibility of setting the node/agents with a respawn property during launch, which means that, when the framework detects that they are not working, it will try to initialize them again, registering the topics and services once again. Thus, it is critical to ensure which agents can be restarted, and if they should recover previous states (e.g., a ROS level sensor driver can restarted and publish data without any issue, while the mission management agent should try to find which task was performing and evaluate the state). This also means that agents relying on data topics will also reconnect automatically to receive published messages, increasing the robustness of the system.

Although several of the agents may recover from an unwanted shutdown due to failure, like those acting as high level sensor drivers, or data processors that do not perform/decide on tasks for the system (data fusion/localization agent, image processing part of the camera agent, and so on), many agents perform tasks that require specific security operations if they become unavailable. The most common approach is for the dependent agents to delay entering the secure operation mode to see if the relevant agent respawns, and keep operating as normal if possible. If normal operation is impossible, the agent will wait in a safe operation mode if possible (i.e., do nothing), delaying the security operation. The main difference between the safe and the security operation modes lies in that, while the safe wait operation implies doing nothing in most cases, the security operation mode cases to abort normal operation and commonly prepare for a restart of the task (e.g., if the mission manager agent fails to recover properly, the avionics agent will start a return-to-home manoeuvre to a fixed safe point).

## 5. Experimental Methods and Results

According to the inputs from stakeholders in the open air basin sampling process, the system was designed considering two different hardware setups to solve the sampling problem: the quick analysis option, deploying a commercial OTS (off-the-shelf) multiparametric sensor; and the alternative of deploying a sample capture device. The first option presents an interesting trade-off between economic cost and simplicity: this kind of sensor is commercially available, widely supported, and integrated into several PC compatible environments, and being a fixed mass, its deployment would prove to be much simpler from an avionic control and operation point of view.

Using a sample capture device required designing and developing a new hardware component. The economic cost of deploying such a device is negligible (once developed) when compared with the parametric sensor, but also presents some weaknesses, namely, most of them related to the fact that the mass will vary and must be submerged, producing a high complexity problem in terms of flight and stability control. Note that, beyond this issue, the sample capturing approach presents multiple advantages beyond its reduced costs, as the analysis will be performed and supervised in a laboratory, being then more accurate, and available when special requirements are needed (keeping samples, processing them before analytics, and so on).

The collector probe developed can be seen in Figure 7. The capacity of the probe caps at ~0.4 l, as any fluid over this volume will not be locked under the buoyancy valve, and thus will be drained out after removing the probe from the basin. The electromagnetic valve, seen in Figure 7a with a release button for testing, operates under negative logic, so loss of current will not translate into losing the sample (dropping a rigid body with a total mass around 0.5 kg during flight would be unsafe). While the probe is submerged up to the filling ports, just above the level of the ring where the buoyancy valve fits to lock, it will be submitted to several forces. While the upward flotation force will be reduced quickly as the probe fills, the filling currents could induce other disturbances. To minimize them, the entry ports are deployed in the opposite side, ensuring that entry flow velocities compensate each other, inducing a global negligible torque. The addition of air evacuation ports near the top reduce the time required to fill the probe (see Table 1) to a third and reduce the disturbances, as the entry flow velocities become uniform and there is no more shaking due to hydrostatic compression of air still inside.

### 5.1. Experimental Evaluation of Available Localiztion Approaches for the Robotics System

Several localization technologies were chosen to be implemented and tested in controlled flights (see Figure 8) in order to operate in open air environments. Though localization can be considered a closed challenge in many circumstances, with plenty of solutions available, it is still an active field of research, as once constraints are introduced into the system, the said solution tends to reduce their effectiveness. Note that there are several trade-offs to consider when choosing which approaches could prove interesting. Highly experimental and generally unsafe solutions were discarded (e.g., there is no visual based SLAM/CML, Simultaneous Localization and Mapping/Concurrent Mapping and Localization, approach able to guarantee industry level robustness in the relocalization problem), as the risks associated with their immaturity outweigh their possible advantages. High accuracy commercial solutions are available, but they tend to be too expensive, as they target the aerospace industry. For example, GPS-based solutions provide the best results in terms of accuracy, but require deployment of expensive equipment both in the UAV and in-site as part of the infrastructure. Thus, this approach was deployed limitedly as a way to obtain a ground truth reference.

Either sampling operation procedures, be it using the multiparametric sensor or the sample capture probe, require great accuracy when estimating the height. Table 2 shows how neither of the GPS approaches considered (fused inertial-DGPS measurements are used as ground truth for evaluation) produce accurate enough raw measurements to be considered usable (GPS, GPS with EGNOS network SBAS augmentation). The laser range-finder (LRF) altimeter deployed is very accurate in terms of linear distance, but its actual height with respect to the ground cannot be computed without additional orientation data.

The avionics agent present at the FMU uses a low-level extended Kalman Filter [31,32] to fuse data from the EGNOS-enabled GPS with the laser altimeter and its internal IMU sensor. These data are usually the backbone of the flight operations, but rely heavily on GPS measurements. This can prove to be problematic as the availability of the EGNOS enhancement is not guaranteed, as has been widely discussed in literature [33]. So, although the avionics agent can reduce height error with the altimeter, the position estimation in plane coordinates (X and Y) is to be considered unreliable, and it is not accurate enough for tasks like landing in limited areas or delivering the sample collection probe.

To solve the challenges produced by the unreliability and limited planar accuracy of the GPS, two additional approaches were considered, integrating them as part of the localization agent at the robotics subsystem. Firstly, a fiducial marker detector and estimator was introduced. This allows to produce accuracy levels in positioning with errors below an inch in limited known areas using image from an ordinary USB. This approach solves the sample delivery and accurate landing challenges.

With respect to the possibility of a GPS shortage/disruption, a system of beacons based in the TI chip CC2530 was tested [29]. The CC2530 deploys full system-on-chip capabilities for short distance wireless applications. Several tests were performed, with those using only the beacon system producing results in compatible ranges with the previous literature [34,35] when assuming a planar distribution of the beacon set. The beacons were deployed covering the edges of a 30 m per 15 m, spaced each 15 m. As an active field or research and testing, further refinement should be able to provide better results, more in line with newer research like the works in [36]. An additional setup, fusing the results from the RMCB beacons with data from the laser altimeter, was tested, but it has proven challenging to finely tune the EKF parameter to produce significant gains.

### 5.2. Prototype Sampling UAV Platform Testing

A set of short flight tests was performed to test the actual autonomous flight capabilities of the built platform beyond the limited scope of those used to evaluate the accuracy of the localization technologies in an environment with access to DGPS. These flights took place in the actual scenario of the waste water processing plant, in an adjacent area to the basin, given the high experimental nature of the hardware deployed.

In these flights (Figure 9), the UAV would take off while carrying a dummy payload of 1 kg to ensure it is able to deal with the additional weight of the sampling technology that would be transported during actual operation. After gaining height until 16 m, it would travel to a point adjacent to the basin to sample, the closest possible to the final sampling point, without entering in the basin limits, and then proceed to fly over the sampling point. This procedure minimizes the time flying over the basin, and thus reduces risks associated to the UAV sampling platform falling into it, and should all the active and passive safety procedures fail. Once the UAV platform was over the sampling point, it decreased height until being 0.4 m over the target surface, at which point it remained there for a period of 20 seconds. Afterwards, the UAV would return to the safe altitude and then proceed to perform slight different trajectories to return to a landing point.

These experiments (with detailed statistics in Table 3, below) were limited in scope owing to safety concerns and the low maturity of the hardware deployed. They still allowed to demonstrate the autonomous flight capabilities of the platform, largely validating the avionics, communications, and power subsystems, and offering partial validation and useful insight into the robotics and control subsystems. The data obtained through the robotics subsystem were recorded and studied through available ROS tools, while the data provided by the avionics subsystem were accessed through a flight reporting utility [37].

## 6. Conclusions

This work describes the architecture and prototype core components designed and built of a solution to automatize the sampling tasks of open air basins and tanks in a wastewater treatment plant. The solution described is conceptually based in operating as if there was a sampling network composed of hundreds of sensors for each basin, which can sample almost any point of it. To operate this virtual network, an actual network of robotized UAV platform equipped with sensors was designed. To build the core managing components of the system and integrate a sample prototype UAV built, a multi-agent system paradigm was used, implementing most of it over an ROS framework.

Using the MAS paradigm with the ROS framework in tandem optimizes the features of each other, maximizing modularity and robustness from a design point of view, and upgradeability and ease of integration and maintenance from a technical point of view. These feature prove to be great advances over the authors’ previous related works [25]. An additional advantage of this system is the scalability, as with proper care, introducing new UAVs into the sampling network is a matter of replicating the required hardware and deploying new instances of the relevant software, which can be then easily configured thanks to the MAS design principles and the reconfiguration capabilities of ROS. At the same time, as long as the MAS specifications are complied with, replacing any software components is trivial, as the worst case would simply require building a parser/translator ROS node to interface as an agent. This contrasts sharply with the majority of the related work studied, where the most frequent approach is built upon a commercially available platform, depending as much as possible on supplier provided technology, thus relying on technologies that cannot be properly maintained or fitted without external dependencies.

With respect to the sampling process itself, a prototype sampling probe for the scenario based on sample capture was built, as it presents the highest potential of both cases: the hardware costs are lower and the analysis would be performed in a laboratory under human supervision and validation. Other works with similar capabilities, like the work of [9], both rely on large-sized platforms, ill-suited for environments with human presence, and yet require human intervention for simple tasks such as delivering the sample. In our case, the probe was tested in water tanks, with a design that optimizes fluid entry speeds, while minimizing torque and disturbances thanks to the entry ports distribution and the airflow outports. To guarantee the safety of the sample, the buoyancy valve deployed showed no losses during the tests under normal circumstances, with similar results achieved for the probe electromagnetic lock.

The robotics subsystem was evaluated in terms of localization, which is usually one of the most complex challenges for platforms with limitations in terms of payload and costs. As a general rule in robotics, as the restrictions in payload and expense per unit become stricter, the localization problem becomes more complex (especially outdoors, where vision based approaches reliability is poor). In the considered system, as the target is to eventually deploy multiple UAVs, restrictions on payload and costs are of the highest level, so a set of alternatives approaches was evaluated. To validate them, accurate DGPS data captured concurrently were used and processed off-line. Conventional SBAS-enhanced GPS provided enough accuracy for low-risk sampling operations, but it was not able to provide enough accuracy to support precise landing and probe delivery operations (beyond risks associated with using an inherently unreliable technology). As alternatives, an RCMB beacon-based system and a fiducial marker-based approach were tested. The beacon-based system provided results with roughly similar accuracy levels (slightly worse), but could provide additional robustness. The fiducial marker-based approach proved to be the best method to manage accurate operation in small, controlled areas, such as landing and probe delivery.

Given the highly experimental nature of the platform, autonomous tests in the actual scenario dealt only with the flight part; in an area adjacent to the basins to be sampled, similar trajectories to those that will be performed during actual operations were executed and tested. These experiments allowed to validate the avionics subsystem and some capabilities of the robotics subsystem, proving that the platform is capable of autonomous flight.

Future works will primarily deal with a limited set of specific problems not addressed in this work. One of the main challenges will be designing a flight stabilization control able to deal with dynamic mass and varying upwards forces during the submersion and filling operation of the UAV, as the current control architecture would be enough in the case of the using the multiparametric sensor, assuming that some hardware design is used so that it can be introduced into the water with the UAV flying over a safe height. 

Development of this control architecture presents some level of dependency with the localization tasks, as changes to the sensor suite can impact it greatly. Additional evaluation of accurate height sensor able to work over water is required, or specific work modelling the different wastewater distortions and bias on LRF measurements. In terms of robotic controls subsystem, integrated localization joining data from the avionics agent and additional robotics localization approach (beacon network, fiducial markers) was proven to produce accurate and robust enough results to operate, although there is still plenty of margin, especially being an active field of research.

## Figures and Tables

**Figure 1 sensors-20-00358-f001:**
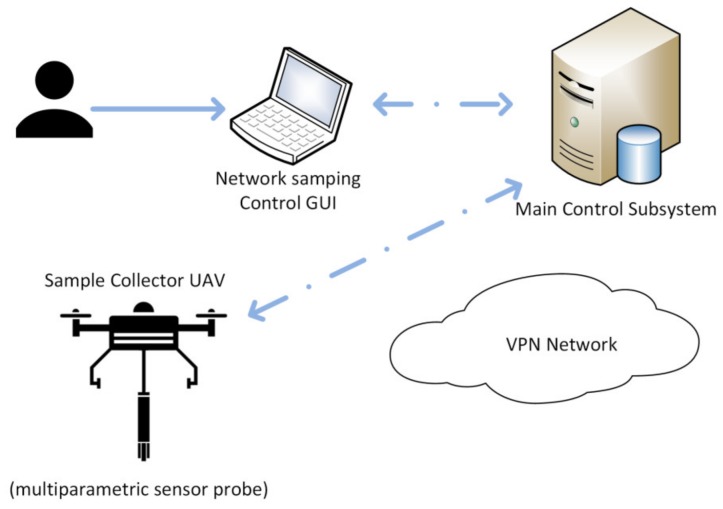
Virtual sensor network using a multiparametric probe. The user interfaces through a remote interface (with a graphical user interface, GUI) with the control system, to designate the sampling operations parameters, all in a Virtual Private Network, VPN. The control system uses these input to relay the mission data to the sampling unmanned aerial vehicle (UAV). In this case, the results can be notified even before the UAV.

**Figure 2 sensors-20-00358-f002:**
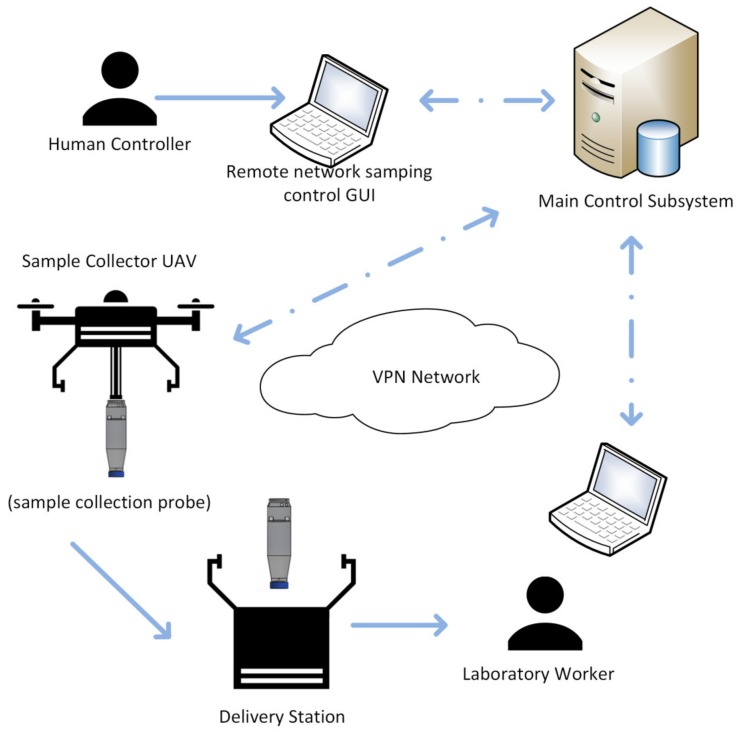
Autonomous UAV-based virtual sensor network for sampling mission when capture of the sample is required. The UAV will submerge the sample collection probe into the water and bring it to a delivery station, where a laboratory worker can retrieve it and run the required analyses.

**Figure 3 sensors-20-00358-f003:**
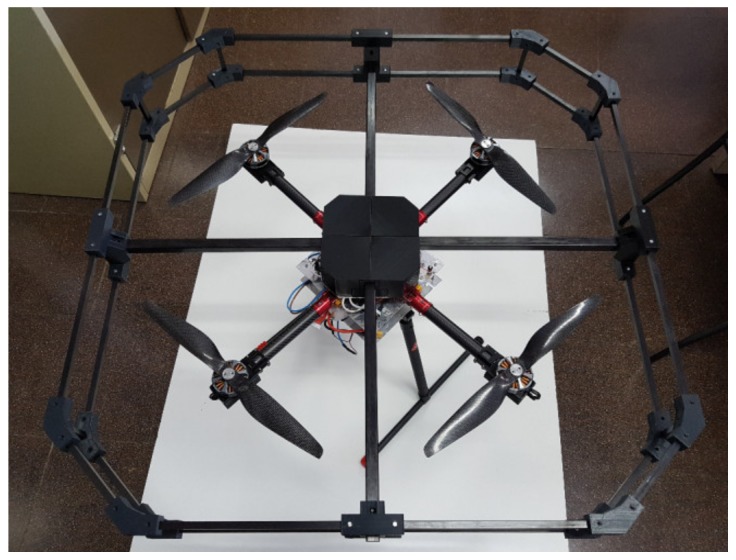
Prototype UAV platform developed as a base for a sample collector UAV part of the UAV sensing network.

**Figure 4 sensors-20-00358-f004:**
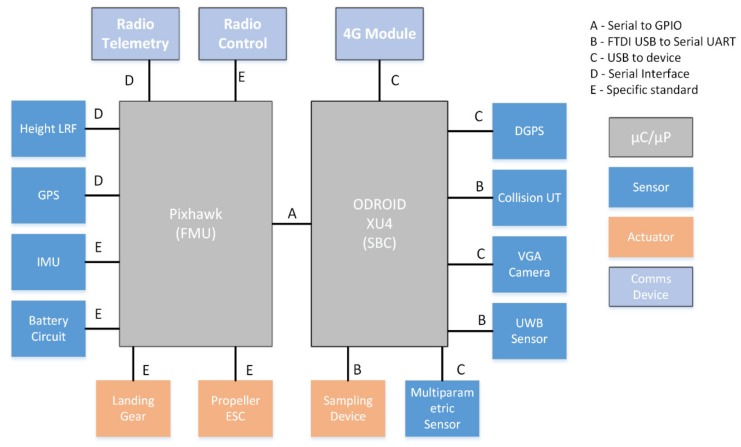
Hardware integration schema for the prototype UAV platform built. Only the sample capture device or the multiparametric probe will be deployed at any given time. DGPS integrated only for testing and evaluation of localization approaches. ESC, electronic speed control; FMU, flight management unit; LRF, laser range finder; SBC, single board computer.

**Figure 5 sensors-20-00358-f005:**
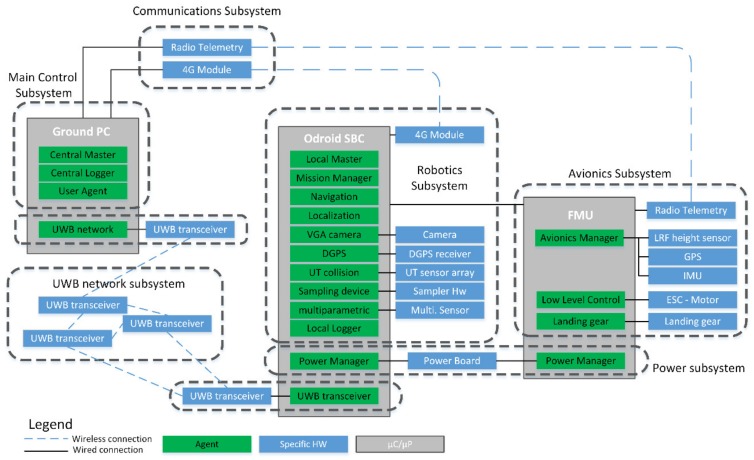
Multi-agent system (MAS) architecture design layout, grouping software, and hardware grouping into subsystems, with detail of hardware (HW) to agent mapping, and agent deployment distribution into the different devices available. Dashed blue lines represent wireless connections; while black solid lines reference physically connected links; and component boxes are coded as green for software, blue for devices, and grey for computing processors.

**Figure 6 sensors-20-00358-f006:**
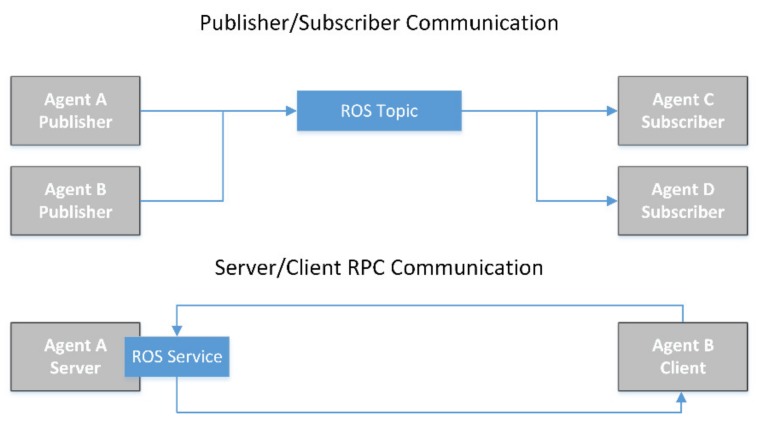
Detail of the two different communications schemas supported by ROS. Top: data are published into topics by publishers (one or more), where any amount of subscribers will receive it. Bottom: under the RPC (remote procedure call) paradigm, each communication instance is initiated by a client who sends a request to a server and receives a response once the request has been processed. Under an ROS framework, this communication (with slight variants) can be implemented either as an ROS service or as an ROS action.

**Figure 7 sensors-20-00358-f007:**
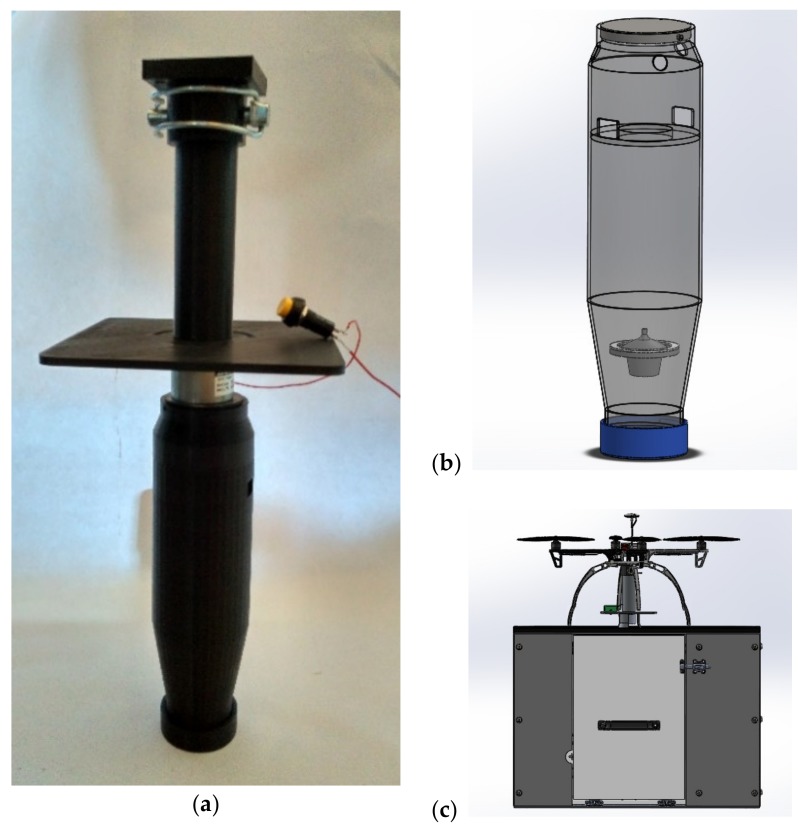
Sample collection probe: (**a**) prototype sample collector build, attached to the electromagnetic valve locking it into the support device. (**b**) CAD design of the probe, with detail of the filling ports, airflow exits, and the buoyancy lock valve. (**c**) Rendering of a UAV landed on a delivery station, with a sample retrieval cabinet.

**Figure 8 sensors-20-00358-f008:**
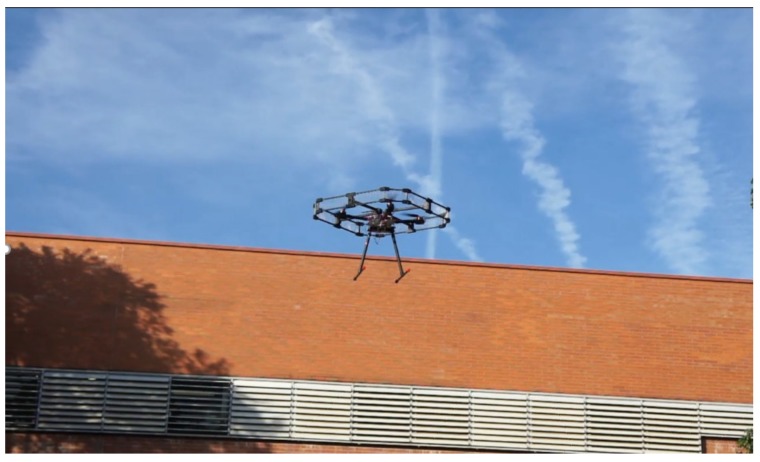
Prototype UAV platform developed during controlled flight tests.

**Figure 9 sensors-20-00358-f009:**
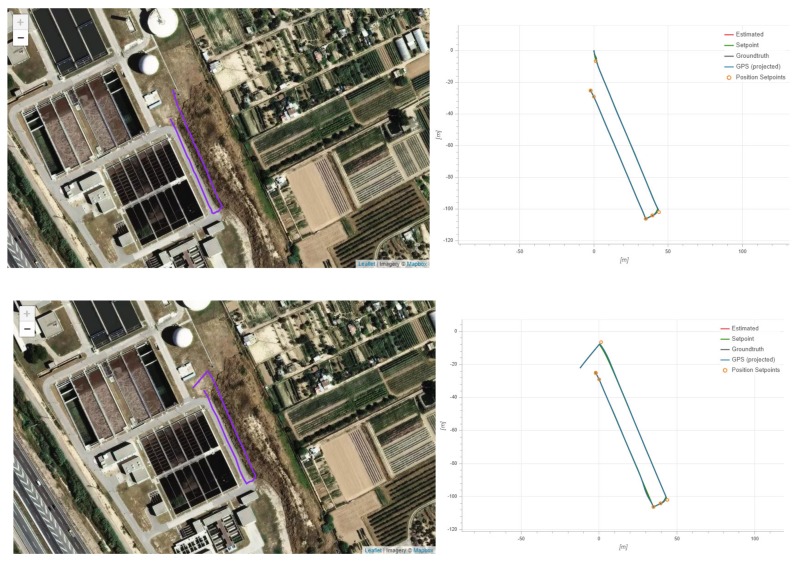
Two of the flights (A and B) performed in the waste water treatment installation with the UAV platform to the autonomous flight capabilities. As no external DGPS was available, the fused position estimation from the avionics system is shown as a ground truth.

**Table 1 sensors-20-00358-t001:** Experimental results with and without airflow exit ports.

Measurement	Avg. Fill Time	σ^2^
No airflow ports	17.5 s	0.16 s^2^
With airflow ports	5.14 s	0.11 s^2^

**Table 2 sensors-20-00358-t002:** Experimental results of the different localization methods evaluated. LRF, laser range finder.

Measurement	Nominal Error		Experimental XY Error ^a^		Experimental Z Error ^a^	
	XY	Z	Average	σ^2^	Average	σ^2^
GPS (raw)	13 m	22 m	6.9 m	3.3 m	9.2 m	7.1 m
GPS (EGNOS, raw)	3 m	8 m	1.2 m	0.5 m	0.9 m	0.3 m
LRF altimeter	^b^	0.05 m	^b^	^b^	0.029 m	0.013 m
Avionics Agent	^b^	^b^	1.1 m	0.27 m	0.019 m	0.034 m
RMCB System (raw) (Range Monte Carlo based)	^b^	^b^	1.17 m	0.5 m	^b^	^b^
Fiducial Marker^c^	^b^	^b^	0.023 m	0.011 m	0.031 m	0.014 m
Loc. Agent (RMCB system, altimeter)	^b^	^b^	1.03 m	0.19 m	0.32 m	0.017 m

^a^ Compared with DGPS results. ^b^ Datum/measurement not available for sensing system. ^c^ Error axes are relative to camera image plane.

**Table 3 sensors-20-00358-t003:** Experimental results of the different localization methods evaluated for flights A and B.

Measurement	Flight A	Flight B
Distance	271.4 m	212.1 m
Max Altitude Difference:	40 m	16 m
Average Speed:	5.3 km/h	4.9 km/h
Max Speed:	11.3 km/h	10.9 km/h
Max Speed Horizontal:	7.5 km/h	7.3 km/h
Max Speed Up:	10.5 km/h	10.7 km/h
Max Speed Down:	5.8 km/h	5.6 km/h
Max Tilt Angle:	21.7 deg	27.3 deg
Max Rotation Speed:	56.9 deg/s	71.1 deg/s

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
