# Peer review of "Design and Implementation of a Virtual Sensor Network for Smart Waste Water Monitoring"

_sensors, 2020, doi:10.3390/s20020358_

Round 1
Reviewer 1 Report
The article is about the generation of a virtual sensor network, through UAV, focused on water monitoring.
The article has academic and mainly technical relevance.
Some suggestions to improve the wording of the text:
- make the objective of the Arctic clearer and more uniform in summary, introduction and conclusion.
- There are few citations in the article and still few of impact.
- Improve the arguments of the advantages and disadvantages of using UAV and boats or other types of robotic vehicles. To discuss.
- The state of the art and technique is poor (discussion and comparisons with results of other works, not enough).
- The data presented in Figure 9 should also be presented as table statistics.
- In conclusion, what is the benefit of this system (more clearly)
- remove "in detail" from the first sentence of the concluding section.
- What is the influence of ROS on the system as a whole? Is there an advantage?
- Use more impactful and current references.
Author Response
Please, read attached document

Reviewer 2 Report
The paper presents an interesting work about the design and implementation of a prototype built such as a solution to automatize the sampling tasks of open air basins and tanks in a wastewater treatment plant. The problem considered by authors is the study to identify the key variables and propose a robotics subsystem to evaluate it.
In general, the paper presents a good work, but it is necessary to solve some details that make that the work carried out is not well understood:
Check the abstract. It doesn’t reflect well the content of the article and is important for online searches.
Review keywords, they are important for the online study search, and it would be advisable to include some more about the algorithm used and the experimental case.
On line 99 you have to remove a "."
The paper does not present an analysis of similar studies, that is, there is no background related to the development carried out on robotized UAVs and applied multi-agent systems. It is true that the paper presents a very detailed developed case in terms of the prototype implemented and evaluated and also references to previous work carried out but does not include an investigation of other similar systems and application of used tecnologies. It would be very convenient to include a background section or something similar.
The paper is well written, correctly structured and follows an appropriated scientific methodology. I recommend accepting this paper and to improve its quality with the suggested changes.
Author Response
Please, read attached document

Reviewer 3 Report
The paper describes the architecture and components of a solution to automatize the sampling tasks of open air basins and tanks in a wastewater
treatment plant. The solution described is based in operating as if there was a sampling network composed of many sensors, replaced by surveillance drones.
Although the article is well written and the idea proposed by the authors is interesting from the point of view of the application,
the auditor does not find particular elements of originality and in-depth analysis of the topics covered regarding the Design and implementation of a virtual sensor network.
In particular, the article contains long and in-depth descriptions of software and hardware tools such as ROS, Odroid, etc. but it lacks information on new results, comparisons with the state of the art of the proposed architectures and performance evaluations of the system here proposed .
Furthermore the authors, proposing the use of MAS strategies to give full solution of the management of a system of surveillance drones, limit themselves to describing their multi-agent solution as a simple use of ROS tools for implementing an architecture of multiagent systems. The same ROS, as a tool, seems to manage a system of agents in a centralized rather than distributed way, even if the authors propose to use a multimaster application to make it distributed. The major concern of the auditor is that the article does not show any behavioral and autonomy strategies for the agents to achieve general objectives that instead seem to have been limited to software parts for managing the hardware tasks of the proposed architecture.
Author Response
Please, read attached document

Round 2
Reviewer 3 Report
The authors have answered sufficiently exhaustively to the reviewers comments